# Toward predictive machine learning for active vision

## Abstract

We develop a comprehensive description of the active inference framework, as proposed by Friston (2010), under a machine-learning compliant perspective. Stemming from a biological inspiration and the auto-encoding principles, a sketch of a cognitive architecture is proposed that should provide ways to implement *estimation-oriented* control policies. Computer simulations illustrate the effectiveness of the approach through a foveated inspection of the input data. The pros and cons of the control policy are analyzed in detail, showing interesting promises in terms of processing compression. Though optimizing future posterior entropy over the actions set is shown enough to attain locally optimal action selection, offline calculation using class-specific saliency maps is shown better for it saves processing costs through saccades pathways pre-processing, with a negligible effect on the recognition/compression rates.

## 1 Motivation

The oculo-motor activity is an essential component of man and animal behavior, subserving most of daily displacements and interactions with objects, devices or people. By moving the gaze with the eyes, the center of sight is constantly and actively moving around during all waking time. The scanning of the visual scene is principally done with high-speed targeted eye movements called saccades (Yarbus (1967)), that sequentially capture local chunks of the visual scene. Though ubiquitous in biology, object recognition through saccades is seldom considered in artificial vision. The reasons are many, of which the existence of high-performance sensors that provide millions of pixels at low cost. Increasingly powerful computing devices are then assigned to compute in parallel those millions of pixels to perform recognition, consuming resources in a brute-force fashion.

The example of animal vision encourages however a different approach towards more parsimonious recognition algorithms. A salient aspect of animal vision is the use of *active* sensing devices, capable of moving around under some degrees of freedom in order to choose a particular viewpoint. The existence of a set of possible sensor movements calls for the development of specific algorithms that should *solve the viewpoint selection problem*. A computer vision program should for instance look back from past experience to see which viewpoint to use to provide the most useful information about a scene. Optimizing the sensor displacements across time may then be a part of computer vision algorithms, in combination with traditional pixel-based operations.

More generally, the idea of viewpoints selection turns out to consider beforehand the computations that need to be done to achieve a certain task. A virtual sensing device should for instance act like a filter that would select which part of the signal should be worth considering, and which part should be bypassed. This may be the case for robots and drones that need to react fast with light and low-power sensing devices. Similarly, in computer vision, Mega-pixel high-resolution images appeals for selective convolution over the images, in order to avoid unnecessary matrix products. Less intuitively, the ever-growing learning databases used in machine learning also suggest an intelligent scanning of the data, in a way that should retain only the critical examples or features, depending on the context, before performing learning on it. Behind the viewpoint selection problem thus lies a feature selection problem, which should rely on a context.

The concept of active vision and/or active perception is present in robotic literature under different acceptances. In Aloimonos et al. (1988), the authors address the case of multi-view image processing

of a scene, i.e. show that some ill-posed object recognition problems become well-posed as soon as several views on the same object are considered. The term was also proposed in Bajcsy (1988) as a roadmap for the development of artificial vision systems, that provides a first interpretation of active vision in the terms of sequential Bayesian estimation, further developed in Najemnik & Geisler (2005); Butko & Movellan (2010); Ahmad & Angela (2013); Potthast et al. (2016).

The active inference paradigm was independently introduced in neuroscience through the work of Friston (2010); Friston et al. (2012). The general setup proposed by Friston and colleagues is that of a general tendency of the brain to counteract surprising and unpredictable sensory events through building generative models that improve their predictions over time and render the world more amenable. This improvement is mainly done through sampling the environment and extracting statistical invariants that are used in return to predict upcoming events. Building a model thus rests on extracting a repertoire of invariants and organizing them so as to process the incoming sensory data efficiently through predictive coding (see Rao & Ballard (1999)). This proposition, gathered under the "Variational Free Energy Minimization" umbrella, is reminiscent of the auto-encoding theory proposed by Hinton & Zemel (1994), but introduces a new perspective on coding for *it formally links dictionary construction from data and (optimal) motor control*. In particular, motor control is here considered as a particular implementation of a *sampling process*, that is at the core of the estimation of a complex posterior distribution.

## 2 ACTIVE INFERENCE

### 2.1 PERCEPTION-DRIVEN CONTROL

The active inference relies on a longstanding history of probabilistic modelling in signal processing and control (see Kalman (1960); Baum & Petrie (1966); Friston et al. (1994)). Put formally, the physical world takes the form of a *generative process* that is the cause of the sensory stream. This process is not visible in itself but is only sensed through a noisy measure process that provides an observation vector $\boldsymbol{x}$. The inference problem consists in estimating the underlying causes of the observation, that rests on a latent state vector $\boldsymbol{z}$ and a control $\boldsymbol{u}$. The question addressed by Friston et al. (2012) is the design a *controller* that outputs a control $\boldsymbol{u}$ from the current $\boldsymbol{z}$ estimate so as to maximize the accuracy of this state estimation process. This is the purpose of a *perception-driven* controller.

Instead of choosing $\boldsymbol{u}$ at random, the general objective of an *active inference* framework is to choose $\boldsymbol{u}$ in a way that should minimize *at best* the current uncertainty about $\boldsymbol{z}$. The knowledge about $\boldsymbol{z}$ can be reflected in a posterior distribution $\rho(\boldsymbol{z})$. The better the knowledge (precision) about a sensory scene, the lower the *entropy* of $\rho$, with :

$$H(\rho) = E_{\boldsymbol{z} \sim \rho}[-\log \rho(\boldsymbol{z})] \tag{1}$$

It is shown in Friston et al. (2012) that minimizing the entropy of the posterior through action can be linked to minimizing the variational free energy attached to the sensory scene. The control $\boldsymbol{u}$ is thus expected to reduce at best the entropy of $\rho$ at each step. This optimal $\boldsymbol{u}$ is not known in advance, because $\boldsymbol{x}$ is only read *after* $\boldsymbol{u}$ has been carried out. Then comes the predictive framework that identifies the effect of $\boldsymbol{u}$ with its most probable outcome, according to the generative model.

If we take a step back, the general formulation of the generative model is that of a feedback control framework, under a discrete Bayesian inference formalism. Given an initial state $\boldsymbol{z}_0$, the prediction rests on two conditional distributions, namely $P(Z|\boldsymbol{u}, \boldsymbol{z}_0)$ – the link dynamics that generates $\boldsymbol{z}$ – and $P(X|\boldsymbol{z}, \boldsymbol{u})$ – the measure process that generates $\boldsymbol{x}$ –. Then, the forthcoming posterior distribution is (Bayes rule) :

$$P(Z|X, \boldsymbol{u}, \boldsymbol{z}_0) = \frac{P(X, Z|\boldsymbol{u}, \boldsymbol{z}_0)}{P(X|\boldsymbol{u}, \boldsymbol{z}_0)} = \frac{P(X|Z, \boldsymbol{u})P(Z|\boldsymbol{u}, \boldsymbol{z}_0)}{\sum_{\boldsymbol{z}'} P(X|\boldsymbol{z}', \boldsymbol{u})P(\boldsymbol{z}'|\boldsymbol{u}, \boldsymbol{z}_0)} \tag{2}$$

so that the forthcoming entropy expectation is :

$$E_X\left[H(\rho)|_{X, \boldsymbol{u}, \boldsymbol{z}_0}\right] = E_X\left[E_Z\left[-\log P(Z|X, \boldsymbol{u}, \boldsymbol{z}_0)\right]\right] \tag{3}$$

and the optimal $\boldsymbol{u}$ is :

$$\hat{\boldsymbol{u}} = \operatorname*{argmin}_{\boldsymbol{u} \in \mathcal{U}} E_X\left[H(\rho)|_{X, \boldsymbol{u}, \boldsymbol{z}_0}\right] \tag{4}$$

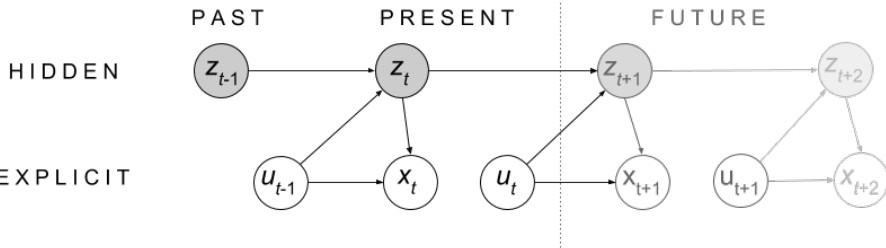

FIGURE 1 – Graphical model (see text)

In practice, the analytic calculations are out of reach (in particular for predicting the next distribution of $\boldsymbol{x}$'s). One thus need to consider an *estimate* $\tilde{\boldsymbol{u}} \simeq \hat{\boldsymbol{u}}$ that should rely on sampling from the generative process to predict the effect of $\boldsymbol{u}$, i.e.

$$\tilde{\boldsymbol{u}} = \underset{\boldsymbol{u}}{\operatorname{argmin}} \frac{1}{N} \sum_{\substack{i=1..N \\ \boldsymbol{x}^{(i)} \sim P(X|\boldsymbol{u},\boldsymbol{z}_0) \\ \boldsymbol{z}^{(i)} \sim P(Z|\boldsymbol{x}^{(i)},\boldsymbol{u},\boldsymbol{z}_0)}} - \log P(\boldsymbol{z}^{(i)}|\boldsymbol{x}^{(i)}, \boldsymbol{u}, \boldsymbol{z}_0) \tag{5}$$

or on an even sharper direct estimation through maximum likelihood estimates (point estimate) :

$$\tilde{\boldsymbol{x}}_{\boldsymbol{u}} = \underset{\boldsymbol{x}}{\operatorname{argmax}} \ P(\boldsymbol{x}|\boldsymbol{u}, \boldsymbol{z}_0) \tag{6}$$

$$\tilde{\boldsymbol{z}}_{\boldsymbol{u}} = \underset{\boldsymbol{z}}{\operatorname{argmax}} \ P(\boldsymbol{z}|\tilde{\boldsymbol{x}}_{\boldsymbol{u}}, \boldsymbol{u}, \boldsymbol{z}_0) \tag{7}$$

$$\tilde{\boldsymbol{u}} = \underset{\boldsymbol{u}}{\operatorname{argmin}} \ - \log P(\tilde{\boldsymbol{z}}_{\boldsymbol{u}}|\tilde{\boldsymbol{x}}_{\boldsymbol{u}}, \boldsymbol{u}, \boldsymbol{z}_0) \tag{8}$$

This operation can be repeated in a sequence, where the actual control $\boldsymbol{u} = \tilde{\boldsymbol{u}}$ is followed by reading the actual observation $\boldsymbol{x}$, which in turn allows to update the actual posterior distribution over the $\boldsymbol{z}$'s. This updated posterior becomes the prior of the next decision step, i.e. $\boldsymbol{z}_0' \sim P(Z|\boldsymbol{x}, \boldsymbol{u}, \boldsymbol{z}_0)$ so that a new control $\boldsymbol{u}'$ can be carried out, etc.

If we denote $T$ the final step of the process, with $\boldsymbol{u}_{0:T-1}$ the actual sequence of controls and $\boldsymbol{x}_{1:T}$ the actual sequence of observations, the final posterior estimate becomes $P(Z_{1:T}|\boldsymbol{x}_{1:T}, \boldsymbol{u}_{0:T-1}, \boldsymbol{z}_0)$, which complies with a Partially Observed Markov Decision Process (POMDP) estimation (see fig. 1), whose policy would have been defined by the entropy minimization principles defined above, precisely to facilitate the estimation process. The active inference framework thus appears as a *scene understanding oriented policy* (it has no other purpose than facilitate estimation).

## 2.2 ACTIVE VISION

The logic behind active vision is that of an external visual scene $\mathcal{X}$ that is never disclosed in full, but only sensed under a particular view $\boldsymbol{x}$ under sensor orientation $\boldsymbol{u}$ (like it is the case in foveated vision). Knowing that $\boldsymbol{z}$ is invariant to changing the sensor position $\boldsymbol{u}$, uncovering $\boldsymbol{z}$ should rest on collecting sensory patches $\boldsymbol{x}$'s through changing $\boldsymbol{u}$ (sensor orientation) across time in order to refine $\boldsymbol{z}$'s estimation. Considering now that a certain prior $\rho_0(\boldsymbol{z})$ has been formed about $\boldsymbol{z}$, choosing $\boldsymbol{u}$ conducts the sight in a region of the visual scene that provides $\boldsymbol{x}$, which in turn allows to refine the estimation of $\boldsymbol{z}$. Each saccade should consolidate a running assumption about the latent state $\boldsymbol{z}$, that may be retained and propagated from step to step, until enough evidence is gathered.

The active vision framework allows many relieving simplification from the general POMDP estimation framework, first in considering that changing $\boldsymbol{u}$ has no effect on the scene constituents, i.e. $P(Z_{t+1}|\boldsymbol{u}, \boldsymbol{z}_t) = P(Z_{t+1}|\boldsymbol{z}_t)$. Then using the *static* assumption, that considers that no significant change should take place in the scene during a saccadic exploration process, i.e. $\forall t, t', \boldsymbol{z}_t = \boldsymbol{z}_{t'} = \boldsymbol{z}$. This finally entails a simplified chaining of the posterior estimation :

$$P(Z|\boldsymbol{x}_{1:t+1}, \boldsymbol{u}_{0:t}, \boldsymbol{z}_0) = \frac{P(\boldsymbol{x}_{t+1}|Z, \boldsymbol{u}_t)P(Z|\boldsymbol{x}_{1:t}, \boldsymbol{u}_{0:t-1}, \boldsymbol{z}_0)}{\sum_{\boldsymbol{z}'} P(\boldsymbol{x}_{t+1}|\boldsymbol{z}', \boldsymbol{u}_t)P(\boldsymbol{z}'|\boldsymbol{x}_{1:t}, \boldsymbol{u}_{0:t-1}, \boldsymbol{z}_0)} \tag{9}$$

issuing a final estimate $P(Z|\boldsymbol{x}_{1:T}, \boldsymbol{u}_{0:T-1}, \boldsymbol{z}_0)$.

**Interpretation** The active inference framework, that is rooted on the auto-encoding theory (Free Energy minimization) and predictive coding, provides a clear roadmap toward an effective implementation in artificial devices. It should rely on three elements, namely :

— a *generative model* $p$ that should predict the next view $\boldsymbol{x}$ under the current guess $\boldsymbol{z}_0$ and viewpoint $\boldsymbol{u}$,

$$p(X|\boldsymbol{u}, \boldsymbol{z}_0) \simeq \sum_{\boldsymbol{z}'} P(X|\boldsymbol{z}', \boldsymbol{u}) P(\boldsymbol{z}'|\boldsymbol{u}, \boldsymbol{z}_0)$$

— an *inference model* $q$ that should predict the next posterior $\boldsymbol{z}$ under putative view $\tilde{\boldsymbol{x}}$ and viewpoint $\boldsymbol{u}$, i.e.

$$q(Z|\tilde{\boldsymbol{x}}, \boldsymbol{u}, \boldsymbol{z}_0) \simeq P(Z|\tilde{\boldsymbol{x}}, \boldsymbol{u}, \boldsymbol{z}_0) \quad \text{— see eq. (2) —}$$

(with the link dynamics $P(Z|\boldsymbol{u}, \boldsymbol{z}_0)$ implicitly embedded in both the generative and inference models in the general case),

— and a policy $\pi$ that should use a "two-steps ahead" prediction (next view prediction first and then inference on predicted view) to issue an optimal control $\boldsymbol{u}$ according to either eq. (5) or eqs. (6–8)

Under the computer vision perspective, and considering $\boldsymbol{z} = \boldsymbol{z}_0$ (static scene assumption), each different $\boldsymbol{u}$ corresponds to a different viewpoint over a static image, with a set of generative $\{p_{\boldsymbol{u}}(X|\boldsymbol{z})\}_{\boldsymbol{u} \in \mathcal{U}}$ and inference $\{q_{\boldsymbol{u}}(Z|\boldsymbol{x})\}_{\boldsymbol{u} \in \mathcal{U}}$ models learned systematically for each different viewpoint $\boldsymbol{u}$. Those place-specific weak classifiers contrast with the place-invariant low-level filters used in traditional image processing (see Viola et al. (2003)) and/or with the first layer of convolution filters used in convolutional neural networks.

# 3 IMPLEMENTATION

## 3.1 ALGORITHMS

As a preliminary step here, we suppose the predictive and generative models are trained apart for we can evaluate the properties of the control policy solely. This *model-based* approach to sequential visual field selection is provided in algorithms 1 and 2.

— A significant algorithmic add-on when compared with formulas (6–8) is the use of a *dynamic actions set* : $\mathcal{U}$. At each turn, the new selected action $\tilde{u}$ is drawn off from $\mathcal{U}$, so that the next choice is made over fresh directions that have not yet been explored. This implements the inhibition of return principle stated in Itti & Koch (2001).

— A second algorithmic aspect is the use of a threshold $H_{\text{ref}}$ to stop the evidence accumulation process when enough evidence has been gathered. This threshold is a free parameter of the algorithm that sets whether we privilege a conservative (tight) or optimistic (loose) threshold. The stopping criterion needs to be optimized to arbitrate between resource saving and coding accuracy.

## 3.2 FOVEA-BASED MODEL

In superior vertebrates, two principal tricks are used to minimize sensory resource consumption in scene exploration. The first trick is the foveated retina, that concentrates the photoreceptors at the center of the retina, with a more scarce distribution at the periphery. A foveated retina allows both treating central high spatial frequencies, and peripheral low spatial frequencies at a single glance (i.e process several scales in parallel). The second trick is the sequential saccadic scene exploration, already mentioned, that allows to grab high spatial frequency information where it is necessary (serial processing).

The baseline vision model we propose relies first on learning local foveated views on images. Consistently with Kortum & Geisler (1996); Wang et al. (2003), we restrain here the foveal transformation to its core algorithmic elements, i.e. the local compression of an image according to a particular focus. Our foveal image compression thus rests on a "pyramid" of 2D Haar wavelet coefficients placed at the center of sight. Taking the example of the MNIST database, we first transform

---

**Algorithm 1** Prediction-Based Policy

---

**Require:** $p$ (generator), $q$ (inference), $\rho$ (prior), $\mathcal{U}$ (actions set)
   predict $z \sim \rho$
   $\forall u \in \mathcal{U}$, generate $\tilde{\boldsymbol{x}}_u \sim p(\boldsymbol{x}|z, u)$
   **return** $\tilde{u} = \underset{u \in \mathcal{U}}{\mathrm{argmax}}\, q(z|\tilde{\boldsymbol{x}}_u, u)$

---

**Algorithm 2** Scene Exploration

---

**Require:** $p$ (generator), $q$ (inference), $\rho_0$ (initial prior), $\mathcal{U}$ (actions set)
   $\rho \leftarrow \rho_0$
   **while** $H(\rho) > H_{\mathrm{ref}}$ **do**
      choose : $\tilde{u} \leftarrow$ Prediction-Based Policy$(p, q, \rho, \mathcal{U})$
      read : $\boldsymbol{x}_{\tilde{u}}$
      update : $\forall z, \mathrm{odd}[z] \leftarrow \log q(z|\boldsymbol{x}_{\tilde{u}}, \tilde{u}) + \log \rho(z)$
      $\rho \leftarrow \mathrm{softmax}(\mathrm{odd})$ {*the posterior becomes the prior of the next turn*}
      $\mathcal{U} \leftarrow \mathcal{U} \setminus \{\tilde{u}\}$
   **end while**
   **return** $\rho$

---

the original images according to a 5-levels wavelet decomposition (see figure 2b). We then define a viewpoint $u$ as a set of 3 coordinates $(i, j, h)$, with $i$ the row index, $j$ the column index and $h$ the spatial scale. Each $u$ may correspond to a visual field made of three of wavelet coefficients $\boldsymbol{x}_{i,j,h} \in \mathbb{R}^3$, obtained from an horizontal, a vertical and an oblique filter at location $(i, j)$ and scale $h$. The multiscale visual information $\boldsymbol{x}_{i,j} \in \mathbb{R}^{15}$ available at coordinates $(i, j)$ corresponds to a set of 5 coefficient triplets, namely $\boldsymbol{x}_{i,j} = \{\boldsymbol{x}_{i,j,5}, \boldsymbol{x}_{\lfloor i/2 \rfloor, \lfloor j/2 \rfloor, 4}, \boldsymbol{x}_{\lfloor i/4 \rfloor, \lfloor j/4 \rfloor, 3}, \boldsymbol{x}_{\lfloor i/8 \rfloor, \lfloor j/8 \rfloor, 2}, \boldsymbol{x}_{\lfloor i/16 \rfloor, \lfloor j/16 \rfloor, 1}\}$ (see figure 2c), so that each multiscale visual field owns 15 coefficients (as opposed to 784 pixels in the original image). Fig. 2d displays a reconstructed image from the 4 central viewpoints at coordinates $(7, 7)$, $(7, 8)$ $(8, 7)$ and $(8, 8)$.

A weak generative model is learned for each $u = (i, j, h)$ (making a total of 266 weak models) over 55,000 examples of the MNIST database. For each category $z$ and each gaze orientation $u$, a generative model is built over parameter set $\Theta_{z,u} = (\rho_{z,u}, \boldsymbol{\mu}_{z,u}, \boldsymbol{\Sigma}_{z,u})$, so that $\forall z, u, \tilde{\boldsymbol{x}}_{z,u} \sim \mathcal{B}(\rho_{z,u}) \times \mathcal{N}(\boldsymbol{\mu}_{z,u}, \boldsymbol{\Sigma}_{z,u})$ with $\mathcal{B}$ a Bernouilli distribution and $\mathcal{N}$ a multivariate Gaussian. The Bernouilli reports the case where the coefficient triplet is null in the considered portion of the image (which is quite common in the periphery of the image), which results in discarding the corresponding triplet from the Gaussian moments calculation. Each resulting weak generative model $p(X|z, u)$ is a mixture of Bernouilli-gated Gaussians over the 10 MNIST labels. For the inference model, a posterior can here be calculated explicitly using Bayes rule, i.e. $q(Z|\boldsymbol{x}, u) = \mathrm{softmax} \log p(\boldsymbol{x}|Z, u)$.

The saccade exploration algorithm is an adaptation of algorithm 2. The process starts from a loose assumption based on reading the root wavelet coefficient of the image, from which an initial guess $\rho_0$

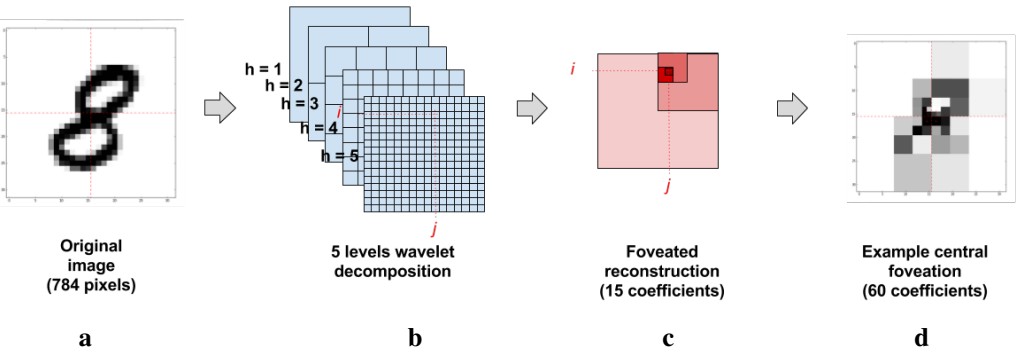

| **Original image (784 pixels)** | **5 levels wavelet decomposition** | **Foveated reconstruction (15 coefficients)** | **Example central foveation (60 coefficients)** |
| :---: | :---: | :---: | :---: |
| **a** | **b** | **c** | **d** |

FIGURE 2 – Foveated image construction.

is formed. Then, each follow-up saccade is calculated on the basis of the final coordinates $(i, j) \in [0, .., 15]^2$, so that the posterior calculation is based on several coefficient triplets. After selecting $(i, j)$, all the corresponding coordinates $(h, i, j)$ are discarded from $\mathcal{U}$ and can not be reused for upcoming posterior estimation (for the final posterior estimate may be consistent with a uniform scan over the wavelet coefficients).

An example of such saccadic image exploration is presented in figure 3**a** over one MNIST sample. The state of the recognition process after one saccade is shown on fig. 3**b**. The next saccade (fig. 3**c**) heads toward a region of the image that is expected to help confirm the guess. The continuing saccade (fig. 3**d**) makes a close-by inspection and the final saccade (fig. 3**e**) allows to reach the posterior entropy threshold, set at $H_{\text{ref}} = 1e^{-4}$ here. The second row shows the accumulation of evidence over the coefficients triplets, with fig. 3**f** showing the posteriors update of different labels and fig. 3**g** showing the posterior entropy update according to the coefficients triplets actually read. Note that several triplets are read for each end-effector position $(i, j)$ (see fig. 2c). There is for instance a total of 5 triplets read out at the initial gaze orientation (**b**), and then 4 triplets read-out for each continuing saccades.

The model provides apparently realistic saccades, for they cover the full range of the image and tend to point over regions that contain class-characteristic pixels. The image reconstruction after 4 saccades allows to visually recognize a "fuzzy" three, while it would not necessarily be the case if the saccades were chosen at random. The observed trajectory illustrates the *guess confirmation* logic that is behind the active vision framework. Every saccade heads toward a region that is supposed to confirm the current hypothesis. This confirmation bias appears counter-intuitive at first sight, for some would expect the eye to head toward places that may *disprove* the assumption (to challenge the current hypothesis). This is actually not the case for the class-confirming regions are more scarce than the class-disproving regions, so that heading toward a class-confirming region may bring more information in the case it would, by surprise, invalidate the initial assumption.

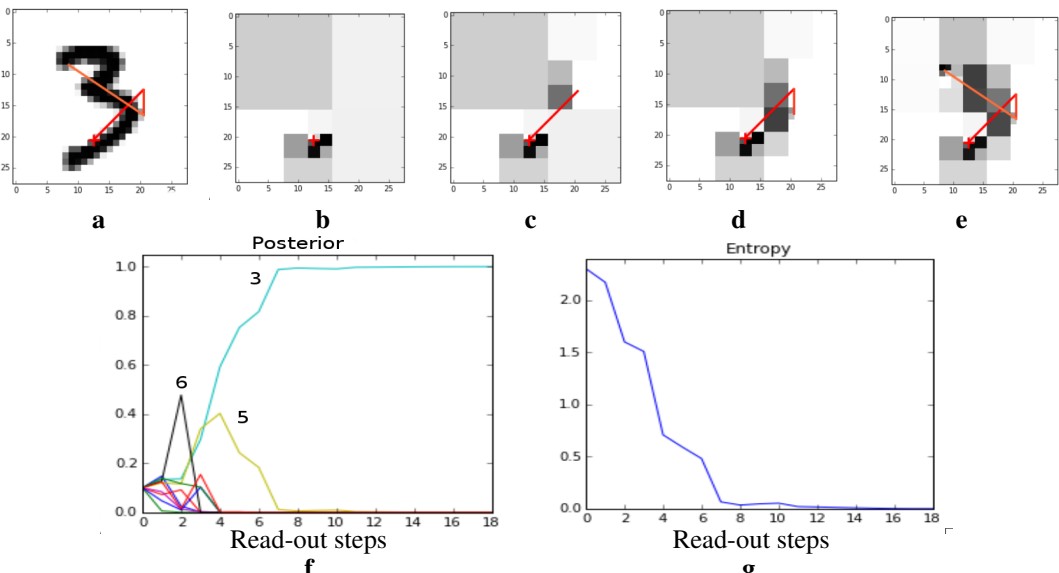

FIGURE 3 – **Image exploration through saccades in the foveated vision model**. **a**. Saccades trajectory over the original image (initial gaze orientation indicated with a red "plus"). **b–e**. Progressive image reconstruction over the course of saccades, with **b** : 5 coefficients triplets + root coefficient (initial gaze orientation), **c** : 9 coefficients triplets + root coefficient (first saccade), **d** : 13 coefficients triplets + root coefficient (second saccade), **e** : 17 coefficients triplets + root coefficient (third saccade) **f**. Posterior update in function of the number of coefficients read-out steps (noting that step 1 stems for the root coefficient and the next steps stem for 3 Haar wavelet coefficients read-out), with one color per category (the numbers over the curves provide the competing labels) **g**. Posterior entropy update in function of the number of read-out steps.

### 3.3 SALIENCY-BASED POLICY

The scaling of the model needs to be addressed when large images are considered. The policy relies on a two-steps ahead prediction (eqs (6–8) and algorithm 1) that scales like $O(|\mathcal{U}|.|\mathcal{Z}|)$ for it predicts the next posterior distribution over the $z$'s for each visual prediction $\boldsymbol{x}_u$. In comparison, parametrized policies are more computationally efficient, allowing for a single draw over the actions set given a context. Luckily, such a parametrized policy is here straightforward to compute. Taking $z_0$ as the initial guess, and noting $\tilde{\boldsymbol{x}}_{u,z_0}$ the visual generative prediction when $z_0$ is assumed under visual orientation $u$, and assuming a uniform prior over the latent states, the process-independent look-ahead posterior is :

$$\rho_{u,z_0}(Z) = \frac{p(\tilde{\boldsymbol{x}}_{u,z_0}|Z, u)}{\sum_{z'} p(\tilde{\boldsymbol{x}}_{u,z_0}|z', u)} \tag{10}$$

providing at each $(u, z_0)$ an offline prediction, namely $\rho_{u,z_0}(z_0)$. Those offline computations provide, for each guess $z_0$, a saliency map over the $u$'s.

Low-level features-based saliency maps date back from Itti & Koch (2001), with many follow-ups and developments in image/video compression (see for instance Wang et al. (2003)). In our case, a saliency map is processed for each guess $z_0$, driving the viewpoint selection regarding $z_0$'s confirmation. Saliency-based policies then allow to define an optimal saccade pathway through the image that follow a sequence of "salient" viewpoints with decreasing saliency (according to the inhibition of return). In our case, the viewpoint selected at step $t$ depends on the current guess $z_t$, with on-the-fly map switch if the guess is revised across the course of saccades.

Examples of such saliency maps are provided in the upper panel of figure 4, for categories 1 to 3. The saliency maps allow to analyze in detail the class-specific locations (that appear brownish) as

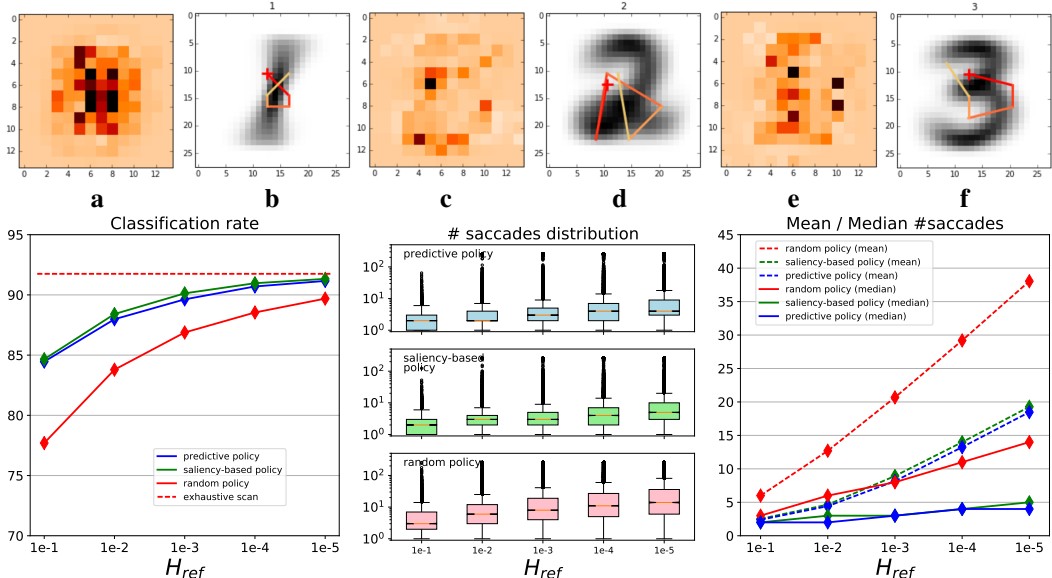

FIGURE 4 – **Saliency based policy – Upper panel : Saliency maps inferred from the model with corresponding saccades trajectory prototypes. a**. Saliency map for latent class "1". **b**. 5-saccades trajectory prototype for latent class "1" (initial position indicated with a red "plus") over class average. **c**. Saliency map for latent class "2". **d**. 5-saccades trajectory prototype for latent class "2" over class average. **e**. Saliency map for latent class "3". **f**. 5-saccades trajectory prototype for latent class "3" over class average. **Lower panel : Policy comparison** (*left*) Average classification rate for the predictive policy, the saliency based policy and a uniform random policy, for different recognition thresholds. The exhaustive scan (baseline) recognition rate is red dashed. (*middle*) Number of saccades distribution for the predictive policy, the saliency-based policy and the random policy. The boxes indicate the first and third quartiles. (*right*) Mean and median number of saccades in function of the recognition threshold for the different considered policies.

opposed to the class-unspecific locations (pale orange to white). First to be noticed is the relative scarceness of the class-specific locations. Those "evidence providing" locations appear, as expected, mutually exclusive from class to class. A small set of saccades is expected to provide most of the classification information while the rest of the image is putatively uninformative (or even counter informative if whitish). A second aspect is that the class-relevant locations are all located in the central part of the images, so there is very few chance for the saccades to explore the periphery of the image where little information is expected to be found. This indicates that the model has captured the essential concentration of class-relevant information in the central part of the images for that particular training set.

The lower part of figure 4 provides an overview of the model behavior in function of the recognition threshold $H_{\text{ref}}$. The original predictive policy is compared to (*i*) the saliency-based policy that selects the saliency map in function of the current guess $z_t$ and (*ii*) a uniform random exploration (choose next viewpoint at random). The classification rates, shown in the leftmost figure, monotonically increase with a decreasing recognition threshold. Considering a 92% recognition rate as the upper bound here (corresponding to an exhaustive decoding made with 266 weak classifiers – a close equivalent of a linear classifier), a near optimal recognition rate is obtained for both the predictive and saliency-based policies for $H_{\text{ref}}$ approaching $1e^{-5}$, while the random policy reveals clearly sub-optimal. A complementary effect is the monotonic increase of the number of saccades with decreasing $H_{\text{ref}}$ shown in the central and rightmost figures. The number of saccades is representative of the recognition difficulty. The distribution of the number of saccades is very skewed in all cases (central figure), with few saccades in most cases, reflecting "peace-of-cake" recognitions, and many saccades more rarely reflecting a "hard-to-reach" recognition. For both the predictive and the saliency-based policies, less than 5 saccades is enough to reach the recognition threshold in more than 50% of the cases (versus about 15 in the random exploration case) for $H_{\text{ref}} = 10^{-5}$.

A strong aspect of the model is thus its capability to do efficient recognition with very few Haar coefficients (and thus very few pixels) in most cases at low computational cost using using either a full predictive policy or pre-processed maps and saccade trajectories. The number of saccades reflects the *processing length* of the scene. For instance, an average number of saccades between 10 and 15 when $H_{\text{ref}} = 1e^{-4}$ corresponds to an average compression of 85-90 % of the data actually processed to recognize a scene. It can be more if the threshold is more optimistic, and less if it is more conservative.

## 4    RELATED WORK AND PERSPECTIVES

Optimizing foveal multi-view image inspection with active vision has been addressed for quite a while in computer vision. Direct policy learning from gradient descent was e.g. proposed in 1991 by Schmidhuber & Huber (1991) using BPTT through a pre-processed forward model. The embedding of active vision in a Bayesian/POMDP evidence accumulation framework dates back from Bajcsy (1988), with a more formal elaboration in Najemnik & Geisler (2005) and Butko & Movellan (2010). It globally complies with the predictive coding framework (Rao & Ballard (1999)) with the predictions from the actual posterior estimate used to evaluate the prediction error and update the posterior. The "pyramidal" focal encoding of images is found in Kortum & Geisler (1996); Wang et al. (2003), with Butko & Movellan (2010) providing a comprehensive overview of a foveated POMDP-based active vision, with examples of visual search in static images using a bank of pre-processed features detectors. Finally, the idea of having many models to identify a scene complies with the weak classifiers evidence accumulation principle (see Viola et al. (2003) and sequels), and generalizes to the multi-view selection in object search and scene recognition Potthast et al. (2016).

Our contribution is twice, for it provides hints toward expressing the view-selection problem in the terms of processing compression under the Free Energy/minimum description length setup (see Hinton & Zemel (1994)), allowing future developments in optimizing convolutional processing (see also Louizos et al. (2017)). A second contribution is a clearer description of the active vision as a two-steps-ahead prediction using the generative model to drive the policy (without policy learning). Though optimizing future posterior entropy over the actions set is shown enough to attain locally optimal action selection, offline calculation using class-specific saliency maps is way better for it saves processing costs by several orders through saccades pathways pre-processing, with a negligible effect on the recognition/compression rates. This may be used for developing active information

search in the case of high dimensionality input data (feature selection problem). The model thus needs to be tested on more challenging computer vision setups, in order to test the exact counterpart of using pre-processed saliency maps with respect to the full predictive case.

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
