# OpenReview forum: "Toward predictive machine learning for active vision"
_ICLR.cc/2018/Conference — Reject_

### Official Review · AnonReviewer2 · 2017-11-26
**Promising work about the analogy between active vision and the free energy principle but currently quite preliminary**

**Rating:** 3
**Confidence:** 4

**Review:**

It is rather difficult to evaluate the manuscript. A large part of the manuscript reviews various papers from the active vision domain and subsequently proposes that this can directly be modeled using Friston’s free energy principle, essentially, by “analogy”, as the authors state. This extends up to page 4. I would argue, that this is quite a stretch, as the free energy principle is essentially blind to the idea of rewards and preferable states such that all tasks are essentially evaluated in terms surprise reduction. This is very much different from large part of the cited classic active vision literature. The authors furthermore introduce a simplification of the setting, i.e. that nothing changes in a scene during saccadic exploration, which is rather unusual for active vision problems.
The authors provide some detail about the actual implementation of their model, section 4, but the in depth details required at ICLR are missing. No comparisons to other gaze selection models or saliency models are given.
Furthermore, the manuscript seems to suggest, that the simulation results are somehow related to human vision as it is stated:
“The model provides apparently realistic saccades, for they cover the full range of the image and tend to point over regions that contain class-characteristic pixels.”
but no actual comparisons or evaluations are provided.

---

> ### Author Response · Authors · 2017-12-05
> **Clarifications on the Free energy setup and various comments**
>
> Sorry but the connection with the encoding/free energy principles is substantial here (it is not a mere analogy). The posterior entropy minimization from action selection directly derives from free energy minimization first principles (see Friston et al 2012 ref in paper that links free energy to posterior entropy minimization). By the way surprise reduction is an objective in itself that can be set up as a reward (an ‘intrinsic’ reward) with possible connections with reinforcement learning “extrinsic” rewards and action optimization.
> Second, the steady state assumption (nothing changes in the scene) is not that uncommon in active vision (see previous comment). Maybe are you puzzled by the difference between a visual scene (the entire image) that doesn’t change (but is covered) and and a view (which is the current perception given viewpoint u) which changes at each saccade?
> Third, a comparison with low-level Itti & Koch saliency models is not relevant here for the MNIST images are not “natural” enough (no texture, flat background etc.). The saliency maps that we build relate to the critical viewpoints where discriminating features are expected (high-level / recognition oriented saliency maps).
> Additional details should be put in the final version regarding the effect of H_ref on classification rates, the comparison with baseline (full image) recognition, random saccades, two-steps ahead predictive policy and pre-processed (high-level saliency maps based) policyn, see :
> https://drive.google.com/open?id=1wFvPUgiN7ekaAaIAQ5KMHgA-H0plttv2
> Last remark : the apparent realism of the saccades relates to the full image coverage but you are right this is not quantified here.

---

> ### Author Response · Authors · 2018-01-12
> **comment on surprise and surprising comments....**
>
> We want to make the point clear that the connection with Friston Free Energy is not a analogy. Sorry to insist but the predictive policy presented in this paper is a direct derivation of Friston active inference principle (minimize Evidence lower bound with action). The trick is to consider the latent state as coding for the entire scene (which only visible in parts). Each partial view thus allows to refine the estimation of z, which in turn makes the next perception less "surprising" (i.e. lowers E_q -log P(x|z, u) for all u). So please reconsider your statement for it is quite deceiving to other reviewers and readers.
> Next the Free Energy principle is a coding optimization principle so it is neither blind or watchful to rewards, it depends how you formulate your problem (this being not the subject though).
> The active inference framework is not "much different" for it has tight relations with the active vision litterature and grounds on the same probabilistic framework (partial observation of a scene, bayesian inference etc..). The static simplification is also extremely classic and present in most cited papers, so it is not unusual at all. This is finally quite a bunch of surprising comments though not critically related to the actual content of the paper!
>
> Next the fovea-based model is given with full implementation detail. Pages 5-6 provide everything needed to reproduce the numerical experiments.
> Last, more comparisons with existing models should indeed be done though there is little room for improvement in the current setting. The missing part/future work  being comparing inhibition of return simplification with trajectory-based optimization.
>
> We also tried to better separate in the new version the review part from the contributions. Apart from the introduction, most of the related work has been pushed to p.8. Pages 3-4 ontain original derivations of the original formula that are not present in the initial papers.

---

### Official Review · AnonReviewer3 · 2017-11-27
**Interesting motivation and formulation**

**Rating:** 5
**Confidence:** 2

**Review:**

This paper introduces a machine learning adaptation of the active inference framework proposed by Friston (2010), and applies it to the task of image classification on MNIST through a foveated inspection of images. It describes a cognitive architecture for the same, and provide analyses in terms of processing compression and "confirmation biases" in the model.
– Active perception, and more specifically recognition through saccades (or viewpoint selection) is an interesting biologically-inspired approach and seems like an intuitive and promising way to improve efficiency. The problem and its potential applications are well motivated.
– The perception-driven control formulation is well-detailed and simple to follow.
– The achieved compression rates are significant and impressive, though additional demonstration of performance on more challenging datasets would have been more compelling

Questions and comments:
– While an 85% compression rate is significant, 88% accuracy on MNIST seems poor. A plot demonstrating the tradeoff of
accuracy for compression (by varying Href or other parameters) would provide a more complete picture of performance. Knowing baseline performance (without active inference) would help put numbers in perspective by providing a performance bound due to modeling choices.
– What does the distribution of number of saccades required per recognition (for a given threshold) look like over the entire dataset, i.e. how many are dead-easy vs difficult?
– Steady state assumption: How can this be relaxed to further generalize to non-static scenes?
– Figure 3 is low resolution and difficult to read.

Post-rebuttal comments:

I have revised my score after considering comments from other reviewers and the revised paper. While the revised version contains more experimental details, the paper in its present form lacks comparisons to other gaze selection and saliency models which are required to put results in context. The paper also contains grammatical errors and is somewhat difficult to understand. Finally, while it proposes an interesting formulation of a well-studied problem, more comparisons and analysis are required to validate the approach.

---

> ### Author Response · Authors · 2017-12-05
> **additional figures**
>
> Thanks for the positive feedback. We intend to put additional figures in the final version, some of them showing the effect of varying Href (speed/accuracy trade-off) as well as the distribution of saccades length. You can find a preview here : https://drive.google.com/open?id=1wFvPUgiN7ekaAaIAQ5KMHgA-H0plttv2
> which should respond most of your questions.
> The steady-state (or static) assumption is quite common in the field for the eyes (or sensor) movements have typically no effect on the environment intrinsic states. The generalization to non-static scenes is straightforward but more difficult to handle (the difficulty lies on building appropriate generative models, i.e. predicting the effect of compound actions and/or external moves on a sensory field made of pixels at reasonable computational cost)

---

### Official Review · AnonReviewer1 · 2017-11-29
**Incremental. Needs editing.**

**Rating:** 3
**Confidence:** 5

**Review:**

In this paper, the authors present a computational framework for the active vision problem. Motivating the study biologically, the authors explain how the control policy can be learned to reduce the entropy of the posterior belief, and present an application (MNIST digit classification) to substantiate their proposal.

I am not convinced about the novelty and contribution of the work. The active vision/sensing problem has been well studied and both the information theory and Bayes risk formulations have already been considered in previous works (see Najemnik and Geisler, 2005; Butko and Movellan, 2010; Ahmad and Yu, 2013).

The paper is also rife with spelling mistakes and grammatical errors and needs a thorough revision. Examples: foveate inspection the data (abstract), may allow to (motivation), tu put it clear (motivation), on contrary to animals retina (footnote 1), minimize at most the current uncertainty (perception-driven control), center an keep (fovea-based implementation), degrade te recognition (outlook and perspective). The citations are in non-standard format (section 1.2: Kalman (1960)).

Overall, I think the paper considers an important problem but the contribution to the state of the art is minimal, and editing highly lacking.

1. J Najemnik and W S Geisler. Optimal eye movement strategies in visual search. Nature, 434(7031):387–91, 2005.
2. N J Butko and J R Movellan. Infomax control of eye movements. IEEE Transactions on Autonomous Mental Development, 2(2):91–107, 2010.
3. S Ahmad and A J Yu. Active sensing as Bayes-optimal sequential decision-making. Uncertainty in Artificial Intelligence, 2013.

---

> ### Author Response · Authors · 2017-12-05
> **Substantial differences with the three provided references**
>
> We agree the 3 kindly provided  references address a similar problem for they use a foveated (partial) view of the scene, and use a sequential evidence accumulation process based on a generative model to uncover the scene. The differences with our work are however substantial. A quite common mistake made in ref. 1 is to define the objective as the one-step forward reconstruction accuracy. This may coincide with the recognition accuracy in special cases, but should fail in most cases (as already pointed out in 2. and 3). In contrast, papers 2 and 3 are about trajectory-based policy optimization with policy learned from a continuous belief vector with function approximators (namely Policy gradient in 2., Monte Carlo/RBF in 3.).
> In our case the policy is not learned but directly processed (optimized) from the generative model. The trick is to do a local optimization using a “two-steps ahead” prediction that predicts the next posterior (the effect of the next observation), consistently with Friston’s active inference/predictive coding approach. Another substantial difference is our scene decoding/feature selection approach that contrasts with the standard visual search tasks and provides a link with classical ML setups.
> Those references will be included in the final version with appropriate comments / comparisons.

---

### Decision · Program_Chairs · 2018-01-29
**ICLR 2018 Conference Acceptance Decision**

**Decision:**

Reject

**Comment:**

All 3 reviewers consider the paper insufficiently good, including a post-rebuttal updated score.
All reviewers + anonymous comment find that the paper isn't well-enough situated with the appropriate literature.
Two reviewers cite poor presentation - spelling /grammar errors making hte paper hard to read.
Authors have revised the paper and promise further revisions for final version.